# H-bonded reusable template assisted *para*-selective ketonisation using soft electrophilic vinyl ethers

Arun Maji[1], Amit Dahiya[1], Gang Lu[2], Trisha Bhattacharya[1], Massimo Brochetta[3], Giuseppe Zanoni[3], Peng Liu[2] & Debabrata Maiti[1,3]

In nature, enzymatic pathways generate $C_{aryl}$—C(O) bonds in a site-selective fashion. Synthetically, $C_{aryl}$—C(O) bonds are synthesised in organometallic reactions using pre-functionalized substrate materials. Electrophilic routes are largely limited to electron-rich systems, non-polar medium, and multiple product formations with a limited scope of general application. Herein we disclose a directed *para*-selective ketonisation technique of arenes, overriding electronic bias and structural congestion, in the presence of a polar protic solvent. The concept of hard–soft interaction along with in situ activation techniques is utilised to suppress the competitive routes. Mechanistic pathways are investigated both experimentally and computationally to establish the hypothesis. Synthetic utility of the protocol is highlighted in formal synthesis of drugs, drug cores, and bioactive molecules.

[1] Department of Chemistry, Indian Institute of Technology Bombay, Mumbai 400076, India. [2] Department of Chemistry, University of Pittsburgh, Pittsburgh, PA 15260, USA. [3] Dipartimento di Chimica, Università degli Studi di Pavia, Viale Taramelli 10, 27100 Pavia, Italy. Correspondence and requests for materials should be addressed to G.Z. (email: gz@unipv.it) or to P.L. (email: pengliu@pitt.edu) or to D.M. (email: dmaiti@iitb.ac.in)

Carbon−Carbon bonds constitute the major backbone of organic molecules. Prudent construction of such linkages facilitates structural manipulation and complex total synthesis.[1] Synthetic methodologies, thriving to transform robust C−H bonds into diverse functional motifs, can potentially shift the retrosynthetic paradigm. Recent exercise on C−H bond functionalization prompted us to sketch a generalised route of site-selective C−C bond formation at a distal *para* position of an arene, attenuating structural and electronic constraints. Although statistically such transformations are highly probable, inert nature and minimal reactivity distinction impose significant synthetic challenges to target a particular C−H bond. In nature, *para*-toluene monooxygenase functionalise *para* C−H bonds of toluene distinctively.[2,3] Although synthetic reproduction of such transformation can be attained for biased substrates, it falters in recapitulating the enzymatic efficiency with unbiased and deactivated substrates.

In biosynthetic pathways, ketoacyl-synthase and benzophenone synthase transfer R−C(O) groups to form C−C(O) connectivity.[4–6] Synthetically carbonyl cores can be accessed using organometallic reagents and cross-coupling at the expense of sensitive reagents and prefunctionalized reactants.[7–13] On the contrary, electrophilic substitutions (Friedel–Crafts acylation) are more atom economical yet biased to electron-rich systems and sensitive to substituents. Electronic effect of the substituents often leads to the inseparable mixtures of isomers whereas it mostly fails with electron-deficient systems. Broadly, the need of non-nucleophilic solvent medium further confines the scope.[14–16] In the present work, we intend to circumvent such limitations both in terms of selectivity as well as reactivity of the reagents. Over the last few decades directed C−H activation has offered a promising strategy for superior regioselectivity.[17–39] However, directed carbonyl insertion is mostly explored for *ortho* C−H bonds.[40–44] Expanding the idea to distal *para* positions, spans larger separation and thus tunnels through bulky and strained intermediates, vulnerable to subtle manifold modification.[45–49]

Comprehending the significance of carbonyl scaffold[50] and synthetic challenges, herein we disclose directed *para*-selective ketonisation of arenes by overriding the electronic bias, in the presence of a polar solvent (Fig. 1).

## Results

**Design of template and optimisation.** Initially acylation reaction was chosen as the prototype transformation with a toluene model substrate, appended with the first-generation biphenyl nitrile directing template (Fig. 2). A series of different acylating agents were screened. Interestingly, the usage of protic solvent under elevated temperature consumed electrophilic acid chlorides and anhydrides. Therefore, selection of a compatible acylating agent was imperative. In view of competitive nucleophilicity of the solvent and metallacycle, we envisioned to identify a soft masked acylating agent to facilitate interaction with soft metallacycles (Fig. 2). In this regard, first breakthrough was obtained with ethyl

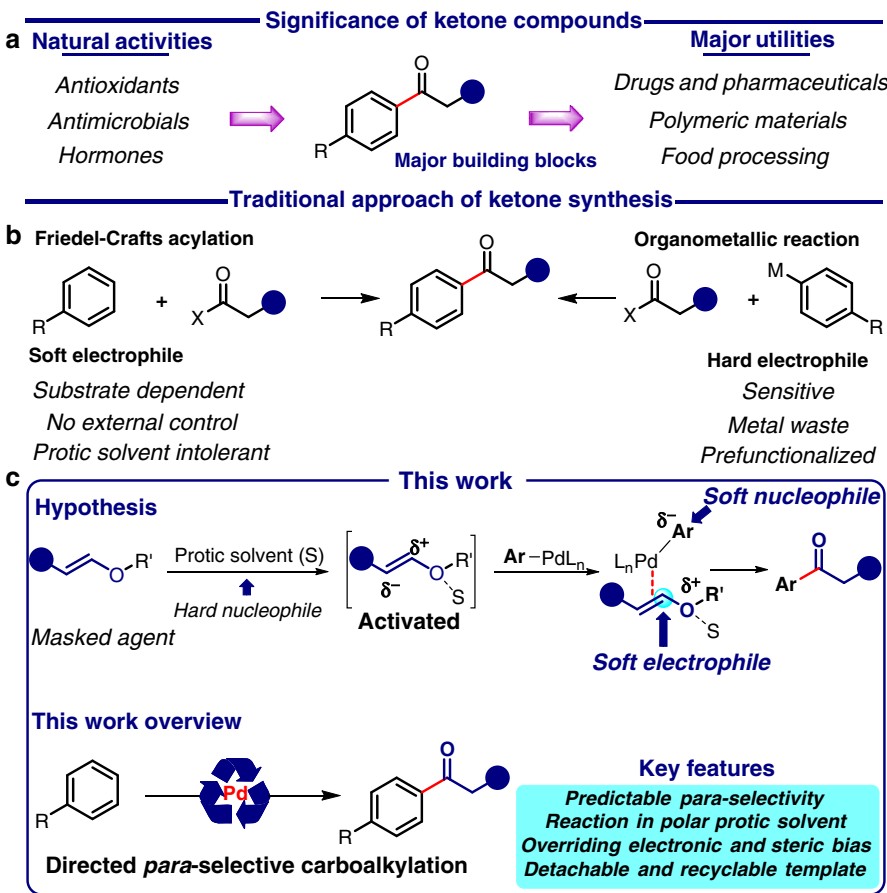

**Fig. 1** Overview of the work. **a** Diverse functions of ketones. **b** Classic synthetic routes for ketone synthesis and its drawbacks. **c** Mechanistic hypothesis for generalised approach and key outline of the work

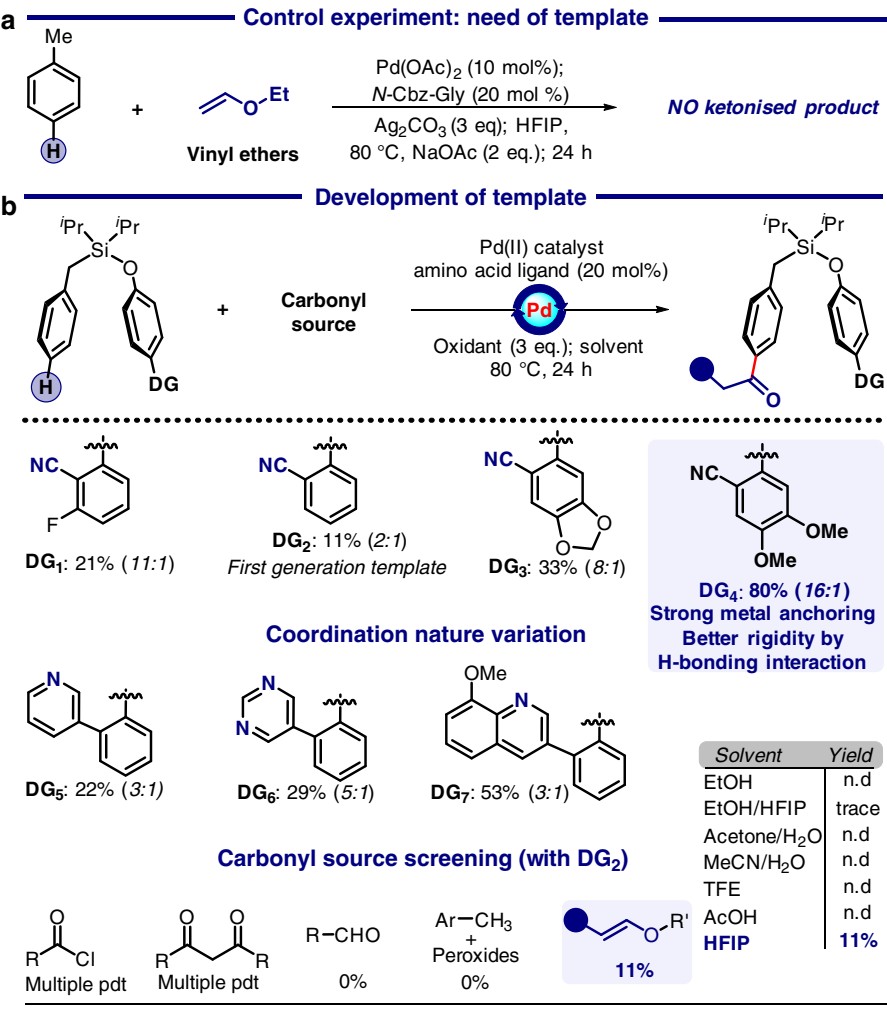

**Fig. 2** Development of *para*-ketonisation reaction. **a** Significance of the directing group. **b** Screening of the reaction parameters

vinyl ether in the presence of catalytic Pd(OAc)$_2$ and hexa-fluoroisopropanol (HFIP) solvent with an overall yield of 11% and 2:1 *para*-selectivity. Notably, vinyl ethers are electron rich and less reactive and thus less pronounced as cross-coupling partner.[51–66] Additionally, the possibility of linear and branched isomer formation along with an additional hydrolysis step to release the carbonyl unit is noteworthy.[67] However, in stark contrast, vinyl ether worked satisfactorily under the current condition in a one-pot process. To seek better selectivity and yield different directing groups were tested. Replacement of the linear nitrile group by heterocyclic metal coordinating motifs such as pyridine (**DG$_5$**), pyrimidine (**DG$_6$**), and methoxy quinoline systems (**DG$_7$**) improved yield yet compromised the selectivity. Apparently, methoxy quinoline (DG$_7$), due to its increased bulk, destabilises the necessary orientation by pushing the toluene nucleus away exposing the *meta*-C–H bond for reaction. In stark contrast, alteration of the electronic environment of the nitrile-based directing group (**DG$_1$**, **DG$_3$**, and **DG$_4$**) offered significant improvement both in yield and selectivity. A yield of 52% with 11:1 *para*-selectivity was obtained with a second-generation hydrogen-bonded *para*-directing template (**DG$_4$**). In particular, the presence of two methoxy groups triggered facile metal-CN binding offering better yield, whereas template-solvent H-bonding interaction generated optimum rigidity to ensure superior selectivity.[46] Under optimised condition Pd(OAc)$_2$ along with *N*-Cbz-Gly gave 80% yield and 16:1 *para*-selectivity in the

presence of NaOAc and Ag$_2$CO$_3$. Control experiment with a simple toluene substrate under the optimised reaction condition gave a mixture of products with no signature of desired *para*-acylated product formation.[68] Such a phenomenon clearly indicates the significant role of the directing template in selective *para* functionalization.

**Scope of the methodology**. Once optimised, the scope of vinyl ether was tested (Table 1). Both cyclic and acyclic alkenyl ethers offered good yield (**1a**–**1c**, **1g**, **1h**, and **1k**). No competitive product formation was observed for allyl vinyl ether, divinyl poly-ether, and free −OH group (**1d**, **1i**, **1j**, and **1l**). Interestingly, vinyl silane and vinyl borane were found to be compatible (**1e** and **1f**). A number of substituents both aliphatic and aromatic moieties around the vinyl group were tested successfully. Both electron-rich and -deficient arene rings were tolerated under the standard condition (**1u**–**1r** and **1w**–**1z**). Di-substitution vinyl ether led to the formation of corresponding α−di (**1z**) substituted ketonised product.

Following the diversification on alkoxy and vinyl substitu-ents, the scope of arenes was explored (Tables 2–4). With electron-rich systems (Tables 2; **2a**–**2i**) a predictable *para*-selectivity was obtained by virtue of the directing group. Despite the possibility of random electrophilic functionaliza-tion, *para*-ketonised product was obtained in synthetically

**Table 1 The scope of vinyl ether**

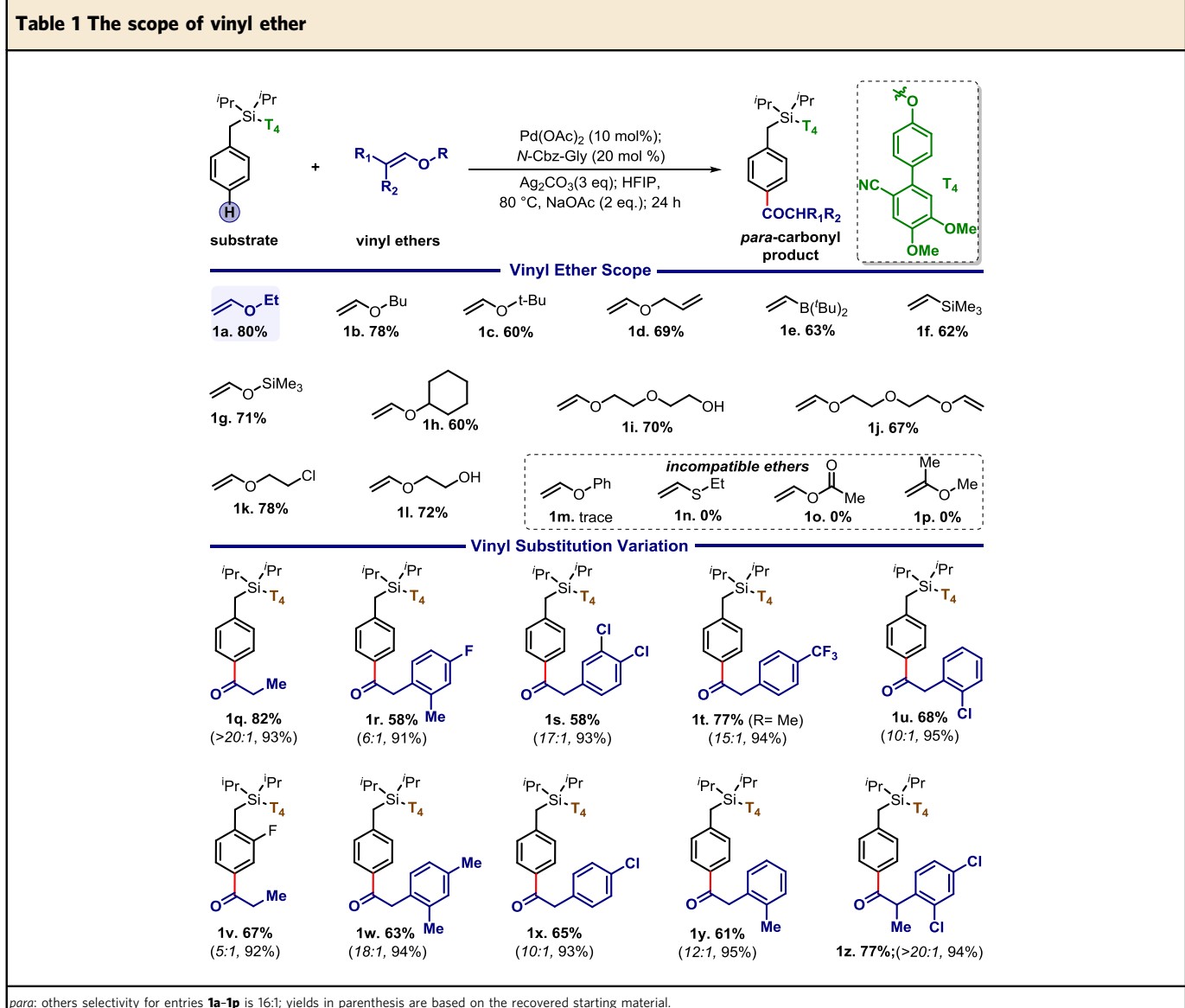

*para*: others selectivity for entries **1a–1p** is 16:1; yields in parenthesis are based on the recovered starting material.

useful yield and selectivity. However, *ortho*-trifluoromethoxy toluene (**2d**) offered a moderate selectivity as compared to methyl (**2b**) substituent. Although the reason of such an anomaly is unclear, it is worth mentioning that −OCF₃ can influence the outcome of a transformation not just by electronic effect but also distorting the planer alignment with the benzene ring which can have a significant impact on the transformation, relied on the appropriate spatial orientation.[69,70] Nevertheless, the current methodology complemented the electrophilic route with excellent *para*-selectivity.

Although electron-deficient systems are not susceptible towards electrophilic substitution reaction, *para*-ketonisation generated the desired product with excellent yield and selectivity (Table 3: **3a–3k**). Poly-halo compounds, specially poly-fluoro, which is having significant medicinal values can be functionalized using the current protocol. Notably, comparable results for both the electron-rich and electron-deficient arenes re-establish the prominent influence of the directing template over other paraphernalia.

Arenes with both electron-rich and electron-deficient substituents were also found to be compatible (Table 4). T₄ template

played the key role to dictate the selectivity. Substrates with benzylic substitution (**4k–4m**) underwent mono *para*-acylation successfully (**4m**).

**Experimental mechanistic evidences.** Following the scope of the reaction, a series of control experiments were conducted to gain better insight of the mechanism (Fig. 3). As the vinyl moiety of ether rearranges to the carbonyl group, we were intrigued to understand the stepwise pathway. NMR titration showed a slow hydrolysis of vinyl ether in the presence of HFIP, generating multiple products including aldehyde which was accelerated upon heating. The control experiment revealed that the hydrolysed product is ineffective as the acylation agent. Upon replacement HFIP by d₂-HFIP or isopropanol, less acidic variant of HFIP, neither decomposition nor the desired product formation was observed. Seemingly, the protonation of the ether by HFIP is responsible for the generation of the reactive intermediate of ketonisation.

During the scope of the reaction thiovinyl ether, unlike vinyl ether, failed completely (Fig. 4a) to deliver the desired carbonyl

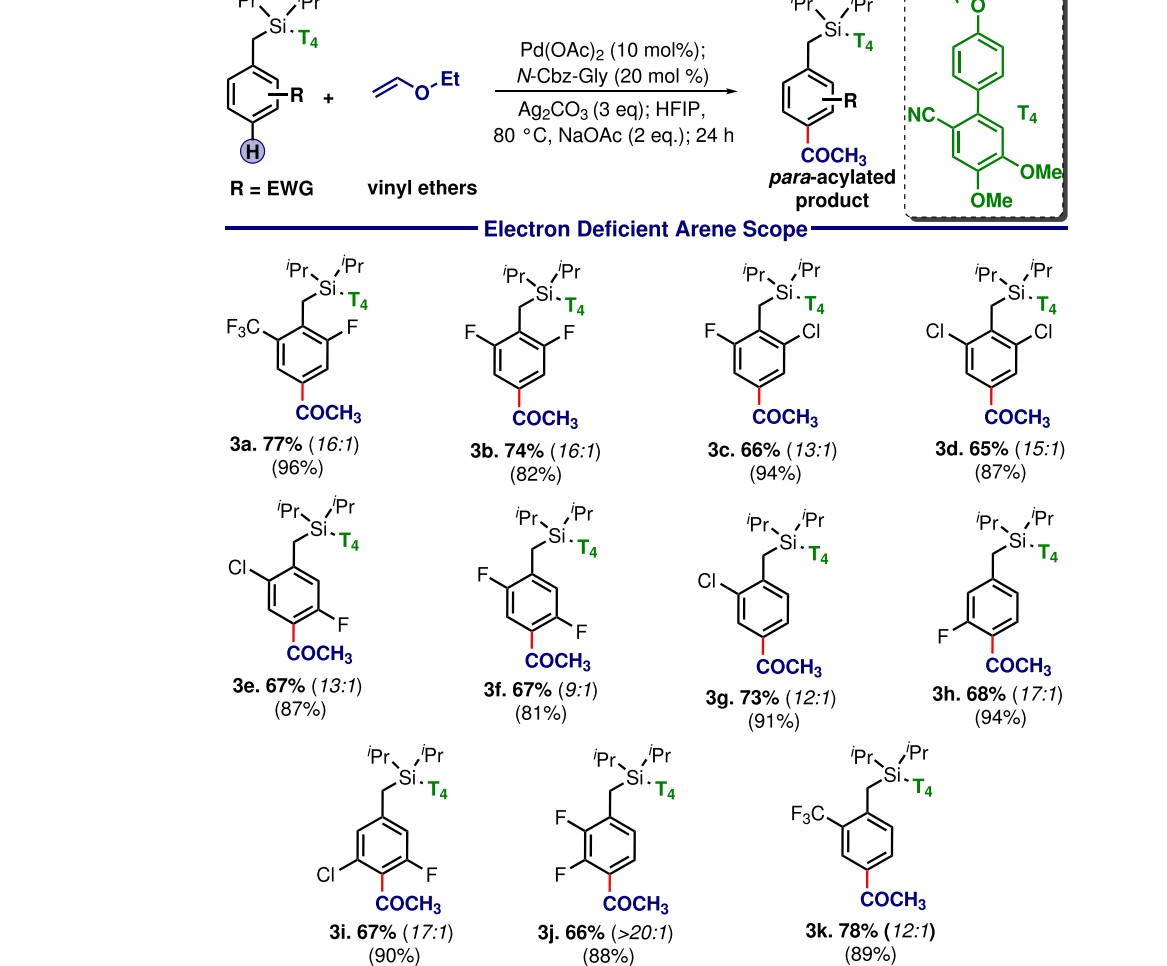

**Table 2 The scope of electron-rich arenes**

**Table 3 Scope with electron-deficient arenes**

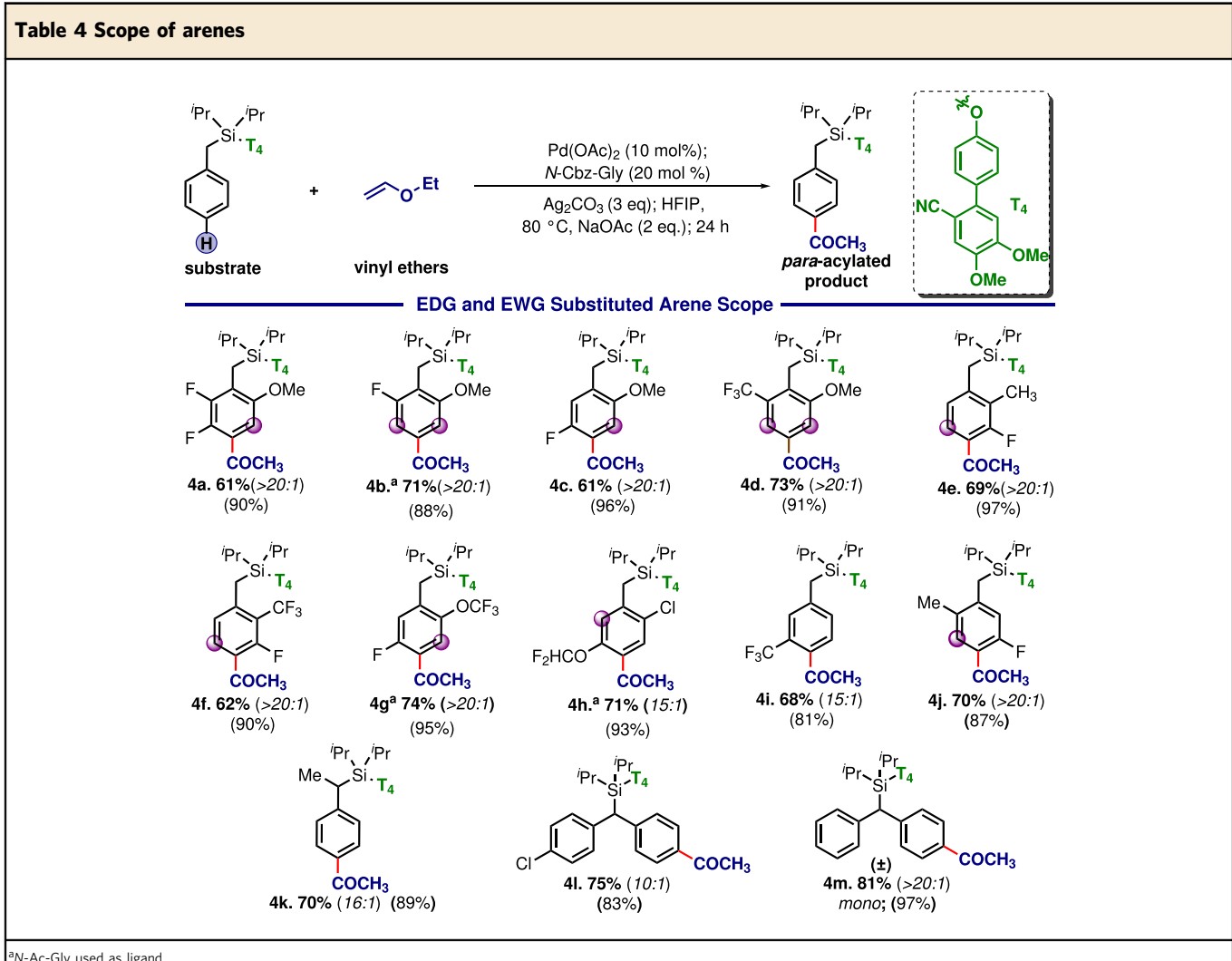

**Table 4 Scope of arenes**

[a]N-Ac-Gly used as ligand

product which further strengthens the hypothesis of protonation by HFIP. Once insertion into the palladacycle, vinyl ether can undergo either nucleophilic pathway (P1) or elimination pathway (P2) of hydrolysis to generate the target molecule. The possibility of both hydrolytic pathways can be rationalised from the reactivity of different alkoxy substituted vinyl ethers (Fig. 4a). Product distribution and kinetic isotope effect study of the deuterated substrates ($k_H/k_D = 3.1$; $P_H/P_D = 3.4$) revealed C−H activation as the rate-limiting step (Fig. 4b).[69]

**DFT calculations and mechanistic cycle**. Based on these mechanistic experiments, a plausible catalytic cycle for *para*-ketonisation was proposed (Fig. 5b). The pathway was evaluated by density functional theory calculations (Fig. 5a). Initial steps of the *para*-selective ketonisation was found to resemble *para*-selective C−H silylation of **1**.[46] Compared to *para*-C–H palladation, *meta*-C–H palladation is disfavoured due to greater ring strain and distortion of the 15-membered palladacycle in the transition state.[46,68] The *para*-C–H metalation occurs via the CMD mechanism directed by the Si-based T₄-directing group to form palladacycle **5**. Subsequent olefin migratory insertion (**TS1**) requires a relatively low activation free energy of 16.0 kcal/mol. The β-elimination of the benzylic hydrogen (**TS2**) is facile, requiring only 10.4 kcal/mol, to form the Pd hydride species **8**, which upon reductive elimination yields the alkenyl ether product **9**. Finally, hydrolysis of alkenyl ether **9** leads to the desired

product formation via pathway **P1** or **P2**. It is noteworthy that apart from generating the activated vinyl ether, HFIP solvent molecule forms H-bonding with the methoxy group of the template (T₄) which favours the bent geometry of C–H metalation transition state, thus metal binding and improved *para* selectivity.[46]

**Template recovery and applications**. During *para*-ketonisation, the presence of the template (T₄) was essential for improved selectivity and yield, yet its removal is required for further synthetic applications (Fig. 6). Almost a quantitative amount of directing was recovered from the **1q** (96%) along with the formation of (p-tolyl)-1-propanone (**5b**) which was further used for α- functionalization and cyclization. Recovery of T₄ from **4m** led to the formation of mono ketonised benzhydryl cores (**5a**).[68]

## Discussion
Therefore, we have developed a reusable template-assisted *para*-selective ketonisation of toluene derivatives with vinyl ethers in the presence of polar protic HFIP. The protocol allows a broad spectrum of vinyl ether and arenes. Also, it can withstand electron deficiency and steric congestion, which is likely to diminish reactivity significantly. The sequence of activation, insertion, and hydrolysis was experimentally investigated and was further supported by computational studies.

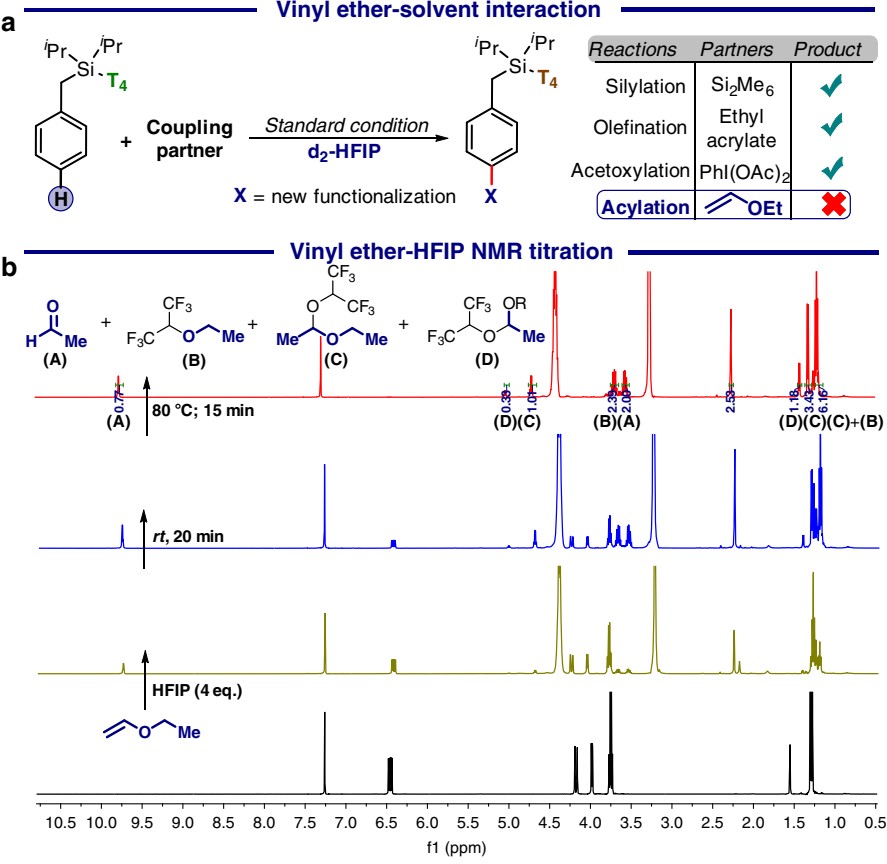

**Fig. 3** Influence of HFIP. **a** Control experiment for vinyl ether–HFIP interaction. **b** NMR study of vinyl ether–HFIP interaction

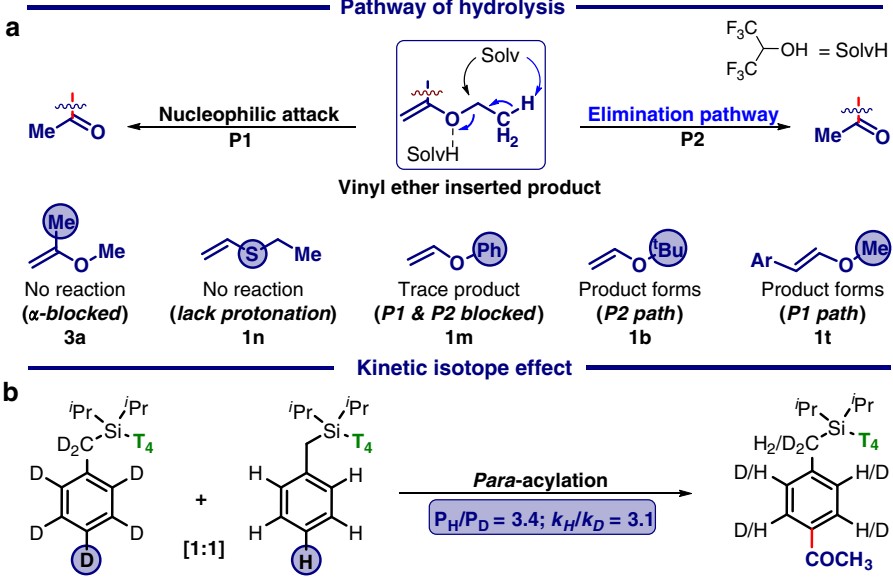

**Fig. 4** Understanding the mechanistic features. **a** Plausible pathways of hydrolysis. **b** Determination of kinetic isotope effect (KIE)

## Methods

**Procedure of *para*-ketonisation.** In an oven-dried screw-capped reaction tube was charged with a magnetic stir-bar, benzylsilyl ether substrate (viscous benzylsilyl ether was weighed first), Pd(OAc)$_2$ (10 mol%), ligand (*N*-CBZ-Gly or *N*-Ac-Gly; 20 mol%), Ag$_2$CO$_3$ (3 eq.) and NaOAc (2 eq.). About 1.2 mL (for 0.1 mmol scale) of 1,1,1,3,3,3-hexafluoro-2-propanol (HFIP) was added followed by vinyl ether (3 eq.). The reaction tube was capped and stirred (900 rpm) on a preheated oil-bath at 80 °C for 24/36 h. Upon completion the mixture was cooled and diluted with EtOAc and filtered through a celite pad. The filtrate was evaporated under reduced pressure and the crude mixture was purified by column chromatography using silica (100–200 mesh size) and petroleum ether/ethyl acetate as the eluent. The selectivity was monitored using $^1$H-NMR signal in the presence of 1,3,5-trimethoxybenzene as an internal standard. The regioselectivity was determined from $^1$H-NMR signals of aromatic region and benzylic position.

In the substrate scope table, selectivity was obtained from $^1$H-NMR.

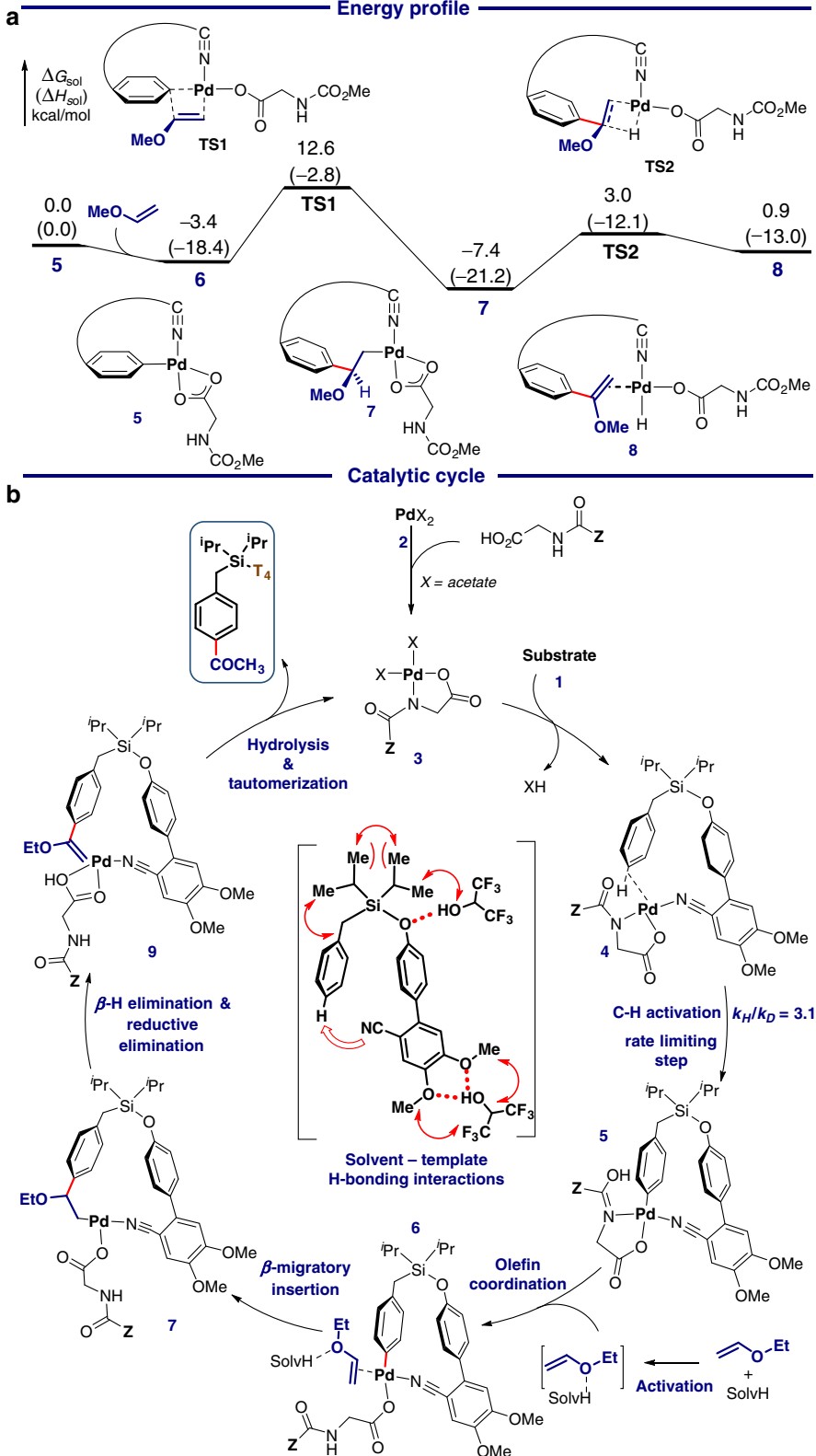

**Fig. 5** Stepwise mechanism of *para*-ketonisation. **a** Energy profile of the *para*-C—H acylation with vinyl methyl ether. Energies are with respect to the palladacycle **5**. See SI for the complete energy profile. Method: M06/SDD-6-311 + G(d,p)/ SMD(HFIP)//B3LYP/SDD-6-31G(d). **b** Plausible catalytic cycle

## Data Availability

The data that support the findings of this study are included in the article and Supplementary Information

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

## Acknowledgements

This activity is supported by research grant from SERB (EMR/2015/000164), India. Financial support received from Lombardy Region and Cariplo Foundation [Regione Lombardia POR FESR 2014-2020/Innovazione e competitività, progetto VIPCAT], Italy, CSIR-India (to A.M.), UGC (to T.B.), NSF (CHE-1654122, to P.L.) and computing time from the Center for Research Computing at the University of Pittsburgh and NSF XSEDE are gratefully acknowledged.

## Author contributions

A.M. conceived the idea and proved it experimentally along with mechanistic control experiments. A.M., A.D., and T.B. performed the substrate scope of the protocol. D.M. supervised the experimental work. M.B. and G.Z. provided analytical reagents. G.L. and P.L. did the computational studies. All the authors contributed to the final version of the manuscript.

## Additional information

**Competing interests:** A provisional patent has been filed on this work.

