## [Peer Review File · Nature Communications]

Reviewer #1 (Remarks to the Author):

The authors report a new directing group for the catalytic para-selective C-H acylation of aromatic rings, along with computational studies that illuminate the origins of regioselectivity and corroborate the proposed reaction mechanism. This process enables access to regioisomers that are inaccessible through conventional Friedel-Crafts reactions and there is an element of conceptual novelty associated with the design of the directing group. Given the importance and topical nature of C-H functionalization, I believe this work would make a fine contribution to Nature Communications.

The authors should include a figure depicting the outcome of a control reaction in the absence of the directing group and contrasting it with the corresponding directed process.

The manuscript contains numerous orthographical errors and should be edited by a native English speaker.

Reviewer #2 (Remarks to the Author):

Maji and coworkers present a para-selective C-H functionalization reaction for arenes. This strategy uses a reusable directing group and therefore represents a notable new pathway for this reaction. Overall, while the concept is interesting and the synthetic scope appears reasonable, the presentation is clearly jammed together in a short length article that doesn't give adequate description of the results. I suggest the authors write out their results/discussion in a longer form, as this work likely would be presented much better with more substantial discussion and analysis.

A couple comments on the mechanistic study:

DFT simulations apparently studied the pathway after C-H activation, whereas C-H activation apparently is rate-limiting. This is a bit odd, and gives no indication of the selectivity of reaction.

The hypothetical role of the solvent is not substantiated, although HFIP is an unusual choice, and appears required for activity. This point needs to be particularly explained, likely simulation can provide real support here.

Reviewer #3 (Remarks to the Author):

The work described by Liu & Maiti and co-workers reports an elegantly para-selective ketonisation of toluenes with both electron-rich and –deficient substituents using a biphenyl-based nitrile template. Different from the conventionally electrophilic substitution acylation (Friedel-Crafts reaction), the process highlights an one-pot directed site-selective palladation/beta-elimination/hydrolysis sequence to form para-C-C(O) bonds. By using vinyl ethyl ether as a masked acylating reagent to avoid the competitive nucleophilicity of the solvent and palladacycle, acylation reaction can be performed in the polar protic solvent (hexafluoroisopropanol), overriding electronic bias and structural constrains of substrates. For electron-withdrawing groups in the toluene ring, para-ketonisation still runs well to give the target acylation products, albeit being less reactive. Given the novelty and interesting advancement of this work, I recommend publishing of this manuscript after addressing the following concerns.

Problems:

1) For electron-rich arene substrates (Table 2), trifluoromethyl group is an electron-withdrawing group and compound 2c should be placed in Table 3.

2) In Table 2, it seems that 2-trifluoromethoxyl substituents (cpd 2e) leads to the decreased regioselectivity (8:1) compared to 2-methyl group (2b, 16:1). Why? The authors should address this point in the main text and give a reasonable explanation.

3) Mechanistically, the present directed para-acylation reaction is fully different from the non-directed electrophilic palladation. To differentiate from them, a control experiment by removing the directing group T4 from the substrate should be performed and the corresponding C-H acylation result reported in the main text for comparison.

Indian Institute of Technology Bombay
Department of Chemistry, Powai, Mumbai, India
400076.

Debabrata Maiti
Associate Professor
Associate Editor, *The Journal of Organic Chemistry*
Ph: 022-2576-7155
Fax: 022-2576-7152
E-mail: dmaiti@chem.iitb.ac.in
<http://www.chem.iitb.ac.in/people/Faculty/prof/dmaiti.ht>

June 8, 2018

To

Editor-In-Chief,
Nature Communication
The Macmillan Campus
London
United Kingdom

Revision of NCOMMS-18-12110

Hello

Attached (electronically) is a revised submission of **NCOMMS-18-12110**, entitled " **H-Bonded Reusable Template Assisted *Para* Selective Ketonisation Using Soft Electrophilic Vinyl Ether**" with revised manuscript and revised supporting information. Corrections/modifications made in the manuscript and in the supporting information are highlighted in **yellow**.

Reviewer 1.

The comments made by Reviewer 1 have helped us to improve the presentation of the manuscript significantly. **Reviewer has appreciated the work presented in the manuscript and recommended for the publication of the revised manuscript in *Nature Communication*.** The proposed corrections are addressed as follows:

Comment 1: The authors should include a figure depicting the outcome of a control reaction in the absence of the directing group and contrasting it with the corresponding directed process

Response 1: As recommended by the Reviewer, control experiments were conducted in absence of the directing group. Simple toluene was treated under the optimized condition in presence ethyl vinyl ether and the product composition was monitored by GCMS and GC. In GCMS mass signals corresponding to the homo-coupled toluene was obtained. In order to ensure the absence of the ketonised product the reaction

mixture was monitored in GC and compared it with the signal of pure 2-methyl acetophenone and 4-methyl acetophenone. No signal corresponding to any of these two compounds was obtained.

The observation was summarized in the manuscript under “Results and Discussions” as

“Control experiment with simple toluene substrate under the optimized reaction condition gave a mixture of products with no signature of desired para-acylated product formation.⁶⁹ Such a phenomenon clearly indicates the significant role of the directing template in selective para functionalization”.

The schematic presentation of the control experiment in contrast to the template assisted approach is included in the Figure 1.

Experimental details are provided in the supporting information as follows at page 50

Control Experiments:

In order to understand the role of directing group, simple toluene as the substrate was treated under the standard condition and the product composition of the reaction mixture was monitored by GC and GCMS. GCMS data showed the presence of toluene homo-coupled product.

GCMS Data:

GCMS assessment report for peak at 7.2 mins:

Unknown: Scan 422 (7.127 min): DM-TB1-58-2R1.D\data.ms
Compound in Library Factor = -1169

Hit 1 : Benzene, 1-methyl-3-(phenylmethyl)-
C₁₄H₁₄; MF: 664; RMF: 825; Prob 10.8%; CAS: 620-47-3; Lib: mainlib; ID: 124559.

Hit 2 : Benzene, 1-methyl-4-(phenylmethyl)-
C₁₄H₁₄; MF: 660; RMF: 753; Prob 9.14%; CAS: 620-83-7; Lib: replib; ID: 21898.

GCMS assessment report for peak at 7.7 mins:

Unknown: Scan 493 (7.723 min): DM-TB1-58-2R1.D\data.ms
Compound in Library Factor = -816

Hit 1 : 4,4'-Dimethylbiphenyl
C₁₄H₁₄; MF: 732; RMF: 906; Prob 13.6%; CAS: 613-33-2; Lib: replib; ID: 23113.

Hit 2 : 3,3'-Dimethylbiphenyl
C₁₄H₁₄; MF: 732; RMF: 868; Prob 13.6%; CAS: 612-75-9; Lib: mainlib; ID: 134734.

The absence of the ketonised product in the reaction mixture was confirmed by comparing the signal with pure 2- and 4-methyl acetophenone.

GC reference for 2-methyl acetophenone:

Front Signal Results

Retention Time	Area	Area %	Height	Height %
2.432	914420225	85.82	808614033	92.97
4.088	151126306	14.18	61136298	7.03
Totals	1065546531	100.00	869750331	100.00

GC reference for 4-methyl acetophenone:

Front Signal Results

Retention Time	Area	Area %	Height	Height %
2.432	951696917	94.16	854245445	96.21
4.274	58976944	5.84	33633221	3.79
Totals	1010673861	100.00	887878666	100.00

GC data of reaction mixture (direct):

**Front Signal
Results**

Retention Time	Area	Area %	Height	Height %
0.053	1572	0.00	489	0.00
0.207	1977	0.00	678	0.00
0.291	862	0.00	425	0.00
0.372	1385	0.00	618	0.00
0.427	709	0.00	575	0.00
0.563	3109	0.00	727	0.00
0.608	3058	0.00	656	0.00
0.731	1641	0.00	645	0.00
0.811	1022	0.00	627	0.00
1.099	6241	0.00	1165	0.00
1.256	5824	0.00	906	0.00
1.327	2041	0.00	901	0.00
1.550	5691	0.00	688	0.00
2.334	344388651	77.22	259023907	74.71
2.457	6687160	1.50	5961264	1.72
2.532	1932058	0.43	1739967	0.50
2.610	67570347	15.15	68769695	19.84
2.720	3673900	0.82	3197471	0.92
2.868	299027	0.07	41679	0.01
3.063	216176	0.05	57152	0.02
3.238	521150	0.12	426473	0.12
3.355	861025	0.19	533713	0.15
3.464	158355	0.04	77712	0.02
3.569	245914	0.06	181460	0.05
3.687	162815	0.04	31756	0.01
3.848	1602334	0.36	1112035	0.32
3.904	1333199	0.30	676136	0.20
4.106	339757	0.08	124831	0.04
4.171	239127	0.05	57637	0.02
4.317	128589	0.03	87880	0.03
4.385	267392	0.06	219534	0.06
4.473	627470	0.14	343990	0.10
4.530	285060	0.06	158976	0.05

GC data of reaction mixture (diluted with ethyl acetate)

Front Signal Results

Retention Time	Area	Area %	Height	Height %
0.060	1577	0.00	615	0.00
0.153	1267	0.00	523	0.00
0.205	2593	0.00	643	0.00
0.342	2828	0.00	825	0.00
0.450	2864	0.00	714	0.00
0.569	3813	0.00	699	0.00
0.739	3279	0.00	828	0.00
0.886	1638	0.00	668	0.00
1.080	6076	0.00	1138	0.00
1.207	5218	0.00	1284	0.00
1.549	16572	0.00	1104	0.00
1.645	1446	0.00	992	0.00
1.775	9141	0.00	1380	0.00
1.982	3783	0.00	795	0.00
2.239	4471	0.00	665	0.00
2.341	2222	0.00	781	0.00
2.568	873645	0.09	462766	0.13
2.779	988908814	99.66	363365652	99.45
3.072	224695	0.02	58618	0.02
3.223	35501	0.00	14116	0.00
3.284	1879017	0.19	1401416	0.38
3.818	55394	0.01	4786	0.00
4.181	1529	0.00	826	0.00
4.448	12829	0.00	2348	0.00
4.513	8875	0.00	1603	0.00
4.632	2999	0.00	1285	0.00
4.696	3307	0.00	1156	0.00
4.858	14493	0.00	1409	0.00
4.984	5375	0.00	1161	0.00
5.154	12393	0.00	1394	0.00
5.283	4897	0.00	1015	0.00
5.616	8289	0.00	963	0.00
5.702	9504	0.00	1322	0.00

Comment 2: Correction of orthographical errors.

Response 2: As suggested by the Reviewer orthographical errors are corrected to the best of our knowledge.

Reviewer 2.

The suggestions made by Reviewer 2 have helped us to improve the presentation of the data. **Reviewer has appreciated the conceptual novelty and the scope of the current transformation and recommended for the publication of the revised manuscript in *Nature Communication*.** The proposed corrections are addressed as follows:

Comment 1: More details in result and discussion section

Response 1: As mentioned by the Reviewer the experimental results were discussed in more details. Following addition/modifications are made to offer a better presentation.

Page 2: *“Initially acylation reaction was chosen as the prototype transformation with toluene model substrate, appended with the 1st generation biphenyl nitrile directing template (Figure 1). A series of different acylating agents were screened.”*

Page 3: *“To seek better selectivity and yield different directing groups were tested. Replacement of the linear nitrile group by heterocyclic metal coordinating motifs such as pyridine (DG₅), pyrimidine (DG₆) and methoxy quinoline systems (DG₇) improved yield yet compromised the selectivity. Apparently, methoxy quinoline (DG₇), due to its increased bulk, destabilizes the necessary orientation by pushing the toluene nucleus away exposing the meta-C-H bond for reaction. In stark contrast, alteration of the electronic environment of the nitrile-based directing group (DG₁, DG₃ and DG₄) offered significant improvement both in yield and selectivity.”*

Page 3: *“Control experiment with simple toluene substrate under the optimized reaction condition gave a mixture of products with no signature of desired para-acylated product formation.⁶⁹ Such a phenomenon clearly indicates the significant role of the directing template in selective para functionalization.”*

Page 5: *“With electron rich systems (Table 2; 2a-2i) a predictable para selectivity was obtained by virtue of the directing group. Despite the possibility of random electrophilic functionalization, para-ketonised product was obtained in synthetically useful yield and selectivity. However, ortho-trifluoromethoxy toluene (2d) offered a moderate selectivity as compared to methyl (2b) substituent. Although, the reason of such an anomaly is unclear, it is worth mentioning that -OCF₃ can influence the outcome of a transformation not just by electronic effect but also distorting the planer alignment with the benzene ring which can have significant impact on the transformation, relied on the appropriate spatial orientation.^{70,71}”*

Page 6: *“Poly-halo compounds, specially poly-fluoro, which is having significant medicinal values can be functionalized using the current protocol. Notably, comparable results for both the electron rich and electron deficient arenes re-establishes the prominent influence of the directing template over other paraphernalia.”*

Page 8 *“Upon replacement HFIP by d₂-HFIP or isopropanol, less acidic variant of HFIP, neither decomposition nor the desired product formation was observed. Seemingly, the protonation of the ether by HFIP is responsible for the generation of the reactive intermediate of ketonisation.”*

Page 8 “During the scope of the reaction thiovinyl ether, unlike vinyl ether, failed completely (Figure 3a) to deliver the desired carbonyl product which further strengthens the hypothesis of protonation by HFIP. Once insertion into the palladacycle, vinyl ether can undergo either nucleophilic pathway (P1) or elimination pathway (P2) of hydrolysis to generate the target molecule. The possibility of both hydrolytic pathways can be rationalized from the reactivity of different alkoxy substituted vinyl ethers (Figure 3a). Product distribution and kinetic isotope effect study of the deuterated substrates ($k_H/k_D = 3.1$; $P_H/P_D = 3.4$) revealed C–H activation as the rate limiting step (Figure 3b).⁶⁹”

Comment 2: DFT simulations apparently studied the pathway after C-H activation, whereas C-H activation apparently is rate-limiting. This is a bit odd and gives no indication of the selectivity of reaction.

Response 2: The complete energy profile, including the C-H activation step, is provided in the SI. Because the mechanism of the C-H activation step and the origin of *para*- versus *meta*-selectivity have been discussed in detail in our recent computational study on the *para*-selective C-H silylation (*Angew. Chem. Int. Ed.* **2017**, 56, 14903), in the present study, we focused on the subsequent functionalization step after the C-H activation. Our calculations indicated the *para*-C-H palladation is the most favourable, in agreement with the experimental selectivity. Compared to *para*-C-H palladation, the disfavoured *meta*-C-H palladation is due to greater ring strain and distortion of the 15-membered palladacycle in the transition state.

The fore-mentioned facts were also mentioned in the manuscript.

Page 8 “Initial steps of the *para*-selective ketonisation was found to resemble *para*-selective C-H silylation of **1**.⁴⁶ Compared to *para*-C-H palladation, *meta*-C-H palladation is disfavoured due to greater ring strain and distortion of the 15-membered palladacycle in the transition state.^{46,69}”

The following sentences were also included in the supporting information

Page 65 “The energetics of the C–H activation pathway of **1** and the origin of *para*-selectivity was found to follow the same trend as observed during *para*-selective silylation reaction.¹ Following the C-H activation,”

Comment 3: The hypothetical role of the solvent is not substantiated, although HFIP is an unusual choice, and appears required for activity. This point needs to be particularly explained, likely simulation can provide real support here

Response 3: The role of the HFIP solvent in a related *para*-C-H silylation was investigated by DFT calculations in our recent paper (*Angew. Chem. Int. Ed.* **2017**, 56, 14903). We found that the HFIP molecules form hydrogen bonds with the oxygen atoms in the template to stabilize the bent geometry of the template in the C-H metalation transition state, thus promoting the metal binding and improving the selectivity. We expect the same solvent effect on the rate-determining C-H activation step assisted by the same template in the present C-H ketonisation reaction.

In order to highlight this fact following sentence is added to the manuscript

Page 9 “It is noteworthy that apart from generating the activated vinyl ether, HFIP solvent molecule forms H-bonding with the methoxy group of the template (**T4**) which favours the bent geometry of C-H metalation transition state, thus metal binding and improved *para* selectivity.⁴⁶”

Reviewer 3.

The comments made by Reviewer 3 have helped us immensely in the betterment of the manuscript. **Reviewer has appreciated the conceptual novelty and recommended for the publication of the revised manuscript in *Nature Communication*.** The concerns are addressed as follows:

Comment 1: For electron-rich arene substrates (Table 2), trifluoromethyl group is an electron-withdrawing group and compound 2c should be placed in Table 3.

Response 1: As recommended by the Reviewer the entry number **2c** in table 2 (previously) is renumber in table 3 as entry **3k**. Other entries are reassigned and the descriptions were rearranged accordingly in manuscript and supporting information and highlighted in **yellow**.

Comment 2: In Table 2, it seems that 2-trifluoromethoxyl substituents (cpd 2e) leads to the decreased regioselectivity (8:1) compared to 2-methyl group (2b, 16:1). Why? The authors should address this point in the main text and give a reasonable explanation.

Response 2: As mentioned by the Reviewer, entry **2e** (currently **2d**) shows a drop-in selectivity as compared to **2b**. Currently, it is not yet completely understood why such a shift was observed. Apparently, -OCF₃ group, unlike -Me or -OMe, shows different characteristics both electronically and in spatial orientation. Interestingly, OCF₃ group can go out of the plan of the aromatic ring in spatial orientation. In the current protocol where the spatial orientation is a key factor of the reaction can get effected by these kinds of conformational changes. For further clarification two relevant references were included as Ref 70 and 71

70 Böhlm, H. J. *et al.* Fluorine in Medicinal Chemistry. *ChemBioChem* **5**, 637-643, doi:10.1002/cbic.200301023 (2004).

71 Leroux, F. R., Manteau, B., Vors, J. P. & Pazenok, S. Trifluoromethyl ethers – synthesis and properties of an unusual substituent. *Beilstein J. Org. Chem.* **4**, 13, doi:10.3762/bjoc.4.13 (2008).

The above mentioned details are summarized in the manuscript as (page 5)

“However, ortho-trifluoromethoxy toluene (2d) offered a moderate selectivity as compared to methyl (2b) substituent. Although, the reason of such an anomaly is unclear, it is worth mentioning that -OCF₃ can influence the outcome of a transformation not just by electronic effect but also distorting the planer alignment with the benzene ring which can have significant impact on the transformation, relied on the proper spatial orientation.^{70,71}”

Comment 3: Mechanistically, the present directed *para*-acylation reaction is fully different from the non-directed electrophilic palladation. To differentiate from them, a control experiment by removing the directing group T4 from the substrate should be performed and the corresponding C-H acylation result reported in the main text for comparison

Response 3: Following the suggestion of the Reviewer we have conducted the control experiment without direction group on toluene substrate and found the substantial difference in the mode of the reaction. During

the control experiment we did not any ketonised product formation. The product composition was monitored by GC and GCMS. To ensure the conclusion GC data was compared with the signal of pure 2-methyl acetophenone and 4-methyl acetophenone. No signal corresponding to any of these two compounds was obtained.

The fact is summarized in the manuscript under “Results and Discussions” as

“Control experiment with simple toluene substrate under the optimized reaction condition gave a mixture of products with no signature of desired para-acylated product formation.⁶⁹ Such a phenomenon clearly indicates the significant role of the directing template in selective para functionalization”.

The schematic presentation of the control experiment in contrast to the template assisted approach is included in the Figure 1.

Experimental details are provided in the supporting information as follows (page 50)

Control Experiments:

In order to understand the role of directing group, simple toluene as the substrate was treated under the standard condition and the product composition of the reaction mixture was monitored by GC and GCMS. GCMS data showed the presence of toluene homo-coupled product.

GCMS Data:

GCMS assessment report for peak at 7.2 mins:

Unknown: Scan 422 (7.127 min): DM-TB1-58-2R1.D\data.ms
Compound in Library Factor = -1169

Hit 1 : Benzene, 1-methyl-3-(phenylmethyl)-
C₁₄H₁₄; MF: 664; RMF: 825; Prob 10.8%; CAS: 620-47-3; Lib: mainlib; ID: 124559.

Hit 2 : Benzene, 1-methyl-4-(phenylmethyl)-
C₁₄H₁₄; MF: 660; RMF: 753; Prob 9.14%; CAS: 620-83-7; Lib: replib; ID: 21898.

GCMS assessment report for peak at 7.7 mins:

Unknown: Scan 493 (7.723 min): DM-TB1-58-2R1.D\data.ms
Compound in Library Factor = -816

Hit 1 : 4,4'-Dimethylbiphenyl
C₁₄H₁₄; MF: 732; RMF: 906; Prob 13.6%; CAS: 613-33-2; Lib: replib; ID: 23113.

Hit 2 : 3,3'-Dimethylbiphenyl
C₁₄H₁₄; MF: 732; RMF: 868; Prob 13.6%; CAS: 612-75-9; Lib: mainlib; ID: 134734.

The absence of the ketonised product in the reaction mixture was confirmed by comparing the signal with pure 2- and 4-methyl acetophenone.

GC reference for 2-methyl acetophenone:

Front Signal Results

Retention Time	Area	Area %	Height	Height %
2.432	914420225	85.82	808614033	92.97
4.088	151126306	14.18	61136298	7.03

Totals	1065546531	100.00	869750331	100.00
--------	------------	--------	-----------	--------

GC reference for 4-methyl acetophenone:

Front Signal Results

Retention Time	Area	Area %	Height	Height %
2.432	951696917	94.16	854245445	96.21
4.274	58976944	5.84	33633221	3.79

Totals	1010673861	100.00	887878666	100.00
--------	------------	--------	-----------	--------

GC data of reaction mixture (direct):

**Front Signal
Results**

Retention Time	Area	Area %	Height	Height %
0.053	1572	0.00	489	0.00
0.207	1977	0.00	678	0.00
0.291	862	0.00	425	0.00
0.372	1385	0.00	618	0.00
0.427	709	0.00	575	0.00
0.563	3109	0.00	727	0.00
0.608	3058	0.00	656	0.00
0.731	1641	0.00	645	0.00
0.811	1022	0.00	627	0.00
1.099	6241	0.00	1165	0.00
1.256	5824	0.00	906	0.00
1.327	2041	0.00	901	0.00
1.550	5691	0.00	688	0.00
2.334	344388651	77.22	259023907	74.71
2.457	6687160	1.50	5961264	1.72
2.532	1932058	0.43	1739967	0.50
2.610	67570347	15.15	68769695	19.84
2.720	3673900	0.82	3197471	0.92
2.868	299027	0.07	41679	0.01
3.063	216176	0.05	57152	0.02
3.238	521150	0.12	426473	0.12
3.355	861025	0.19	533713	0.15
3.464	158355	0.04	77712	0.02
3.569	245914	0.06	181460	0.05
3.687	162815	0.04	31756	0.01
3.848	1602334	0.36	1112035	0.32
3.904	1333199	0.30	676136	0.20
4.106	339757	0.08	124831	0.04
4.171	239127	0.05	57637	0.02
4.317	128589	0.03	87880	0.03
4.385	267392	0.06	219534	0.06
4.473	627470	0.14	343990	0.10
4.530	285060	0.06	158976	0.05

GC data of reaction mixture (diluted with ethyl acetate)

**Front Signal
Results**

Retention Time	Area	Area %	Height	Height %
0.060	1577	0.00	615	0.00
0.153	1267	0.00	523	0.00
0.205	2593	0.00	643	0.00
0.342	2828	0.00	825	0.00
0.450	2864	0.00	714	0.00
0.569	3813	0.00	699	0.00
0.739	3279	0.00	828	0.00
0.886	1638	0.00	668	0.00
1.080	6076	0.00	1138	0.00
1.207	5218	0.00	1284	0.00
1.549	16572	0.00	1104	0.00
1.645	1446	0.00	992	0.00
1.775	9141	0.00	1380	0.00
1.982	3783	0.00	795	0.00
2.239	4471	0.00	665	0.00
2.341	2222	0.00	781	0.00
2.568	873645	0.09	462766	0.13
2.779	988908814	99.66	363365652	99.45
3.072	224695	0.02	58618	0.02
3.223	35501	0.00	14116	0.00
3.284	1879017	0.19	1401416	0.38
3.818	55394	0.01	4786	0.00
4.181	1529	0.00	826	0.00
4.448	12829	0.00	2348	0.00
4.513	8875	0.00	1603	0.00
4.632	2999	0.00	1285	0.00
4.696	3307	0.00	1156	0.00
4.858	14493	0.00	1409	0.00
4.984	5375	0.00	1161	0.00
5.154	12393	0.00	1394	0.00
5.283	4897	0.00	1015	0.00
5.616	8289	0.00	963	0.00
5.702	9504	0.00	1322	0.00

Based on the contribution made during the revision of the manuscript we have included Mr. Massimo Brochetta and Prof. Giuseppe Zanoni as the co-author of this paper.

I hope Reviewers will find the responses satisfactory and will recommend for the final acceptance in *Nature Communication*.

Best regards,

Debabrata Maiti

encl. (electronically): manuscript, supporting information

Reviewer #3 (Remarks to the Author):

The revision made by the authors is satisfying. OK to publish!